# Contrarian Voter Model under the Influence of an Oscillating Propaganda: Consensus, Bimodal Behavior and Stochastic Resonance

**DOI:** 10.3390/e24081140

**Published:** 2022-08-17

**Authors:** Maria Cecilia Gimenez, Luis Reinaudi, Federico Vazquez

**Affiliations:** 1FaMAF (UNC), IFEG (CONICET), Córdoba X5000HUA, Argentina; 2Facultad de Ciencias Químicas (UNC), INFIQC (CONICET), Córdoba X5000HUA, Argentina; 3Instituto de Cálculo, FCEyN, Universidad de Buenos Aires and Conicet, Intendente Guiraldes 2160, Cero + Infinito, Buenos Aires C1428EGA, Argentina

**Keywords:** opinion dynamics, voter model, noise, periodic field, stochastic resonance

## Abstract

We study the contrarian voter model for opinion formation in a society under the influence of an external oscillating propaganda and stochastic noise. Each agent of the population can hold one of two possible opinions on a given issue—against or in favor—and interacts with its neighbors following either an imitation dynamics (voter behavior) or an anti-alignment dynamics (contrarian behavior): each agent adopts the opinion of a random neighbor with a time-dependent probability p(t), or takes the opposite opinion with probability 1−p(t). The imitation probability p(t) is controlled by the social temperature *T*, and varies in time according to a periodic field that mimics the influence of an external propaganda, so that a voter is more prone to adopt an opinion aligned with the field. We simulate the model in complete graph and in lattices, and find that the system exhibits a rich variety of behaviors as *T* is varied: opinion consensus for T=0, a bimodal behavior for $T<T_{c}$, an oscillatory behavior where the mean opinion oscillates in time with the field for $T>T_{c}$, and full disorder for T≫1. The transition temperature $T_{c}$ vanishes with the population size *N* as $T_{c}≃2/\ln N$ in complete graph. In addition, the distribution of residence times $t_{r}$ in the bimodal phase decays approximately as $t_{r}^{-3/2}$. Within the oscillatory regime, we find a stochastic resonance-like phenomenon at a given temperature $T^{*}$. Furthermore, mean-field analytical results show that the opinion oscillations reach a maximum amplitude at an intermediate temperature, and that exhibit a lag with respect to the field that decreases with *T*.

## 1. Introduction

Agent-based models are a powerful tool used in disciplines such as statistical physics, computer science and mathematics to study different collective social phenomena, including opinion formation. Among them, the voter model (VM) [1,2] is one of the simplest agent-based models for opinion dynamics, in which each agent (voter) of a population can hold one of two possible opinion states (+1 and −1) at a given time, and it is allowed to update its opinion by a social imitation mechanism. The two opinion values could represent a position on a given issue, in favor (+1) or against (−1). In a single iteration step of the dynamics, a randomly chosen voter simply adopts the opinion of a random neighbor. This dynamics is repeated until the population ultimately reaches consensus, with all voters sharing the same opinion; a collective absorbing state where opinions can no longer change. Although this basic formulation of the VM is too simple to properly describe the dynamics of opinions in a real society, it provides a theoretical framework to explore how an imitation process in a population of individuals can lead to opinion consensus [3].

In an attempt to model more realistic scenarios, many variations of the VM have been proposed and studied (see reviews [4,5] and references therein). Some examples include multiple opinion states [6,7,8], individual heterogeneity and stubbornness [9], opinion leaders and zealots [10], and complex interaction topologies that could be static [11,12,13] or evolve in time [14,15,16]. In particular, it is expected that opinion patterns in real life resemble those of coexistence of both opinions, rather than a perfect consensus of a single opinion. One way to achieve coexistence within the VM is by including some contrarian agents, i.e., agents that try to adopt the opposite opinion of their neighbors [17,18,19,20,21,22,23,24,25,26,27,28,29]. In the contrarian voter model (CVM) introduced by Banisch in [27], each agent can behave as a “contrarian” with probability *p*, i.e., adopts the opposite opinion of its interacting partner, or as a “voter” with the complementary probability 1−p, i.e., imitates the opinion of its partner. He showed analytically that there is a transition at a contrarian rate $p^{*}=1/\left(N+1\right)$ in a population of *N* agents on a complete graph (all-to-all interactions), from an ordered phase for $p<p^{*}$ characterized by a bimodal distribution of opinions to a disordered phase for $p>p^{*}$, with a unimodal opinion distribution. In the ordered phase, the population is composed by a large majority holding one opinion (quasi-consensus), while in the disordered phase the number of agents in each opinion fluctuates around N/2. In an earlier study [26], Masuda investigated the VM with a fixed number of agents that have the role of contrarians, in three different versions depending on their interaction partner. We note that this model differs from that of Banisch where each agent can behave as a “contrarian” or as a “voter” (annealed disorder) with probabilities *p* and 1−p, respectively, whereas in the model studied by Masuda each agent has a fixed role, behaves always either as a “contrarian” or as a “voter” (quenched disorder).

The seminal work by Galam [17] was the first to study the influence of a fixed number of contrarians in the majority rule (MR) model. Unlike the VM where interactions are pairwise, in the original MR model agents interact in groups, and take the opinion of the majority. This dynamics eventually leads to full consensus. However, this perfect consensus is removed with the introduction of contrarian agents that adopt the opinion opposite to the majority, i.e., the minority opinion. He found that the system reaches an ordered steady state if the fraction of contrarians *a* is smaller than a threshold value $a_{c}$, and a disordered steady state if $a>a_{c}$. The ordered phase is characterized by a large fraction of the population holding the same opinion (close to full consensus), while in the disordered phase there is a balanced coexistence of both opinion groups, with a similar fraction of agents holding each opinion. The presence of contrarians has also been analyzed within the Sznajd model [20,21,22,23,24,25], where it was found that a contrarian-like effect can spontaneously emerge when a stochastic driving is included in the model [20]. The effects of contrarians have also been investigated in several other opinion models [30,31,32,33,34,35,36].

Another realistic feature at the collective level that was included in opinion models is an external mass media propaganda, which influences public opinion in the preferred direction set by the propaganda [25,37,38,39,40,41]. The case scenario of a propaganda that oscillates in time was considered in several studies [24,25,42,43,44,45,46] where, in analogy with spin systems in statistical physics, the propaganda can be viewed as an external oscillating field that has an influence on the opinion update of every agent. In these works, they observed the emergence of a stochastic resonance (SR) phenomenon, in which there is an optimal level of noise for the population to respond to the external field modulation [47,48]. The work by Kuperman and Zanette [42] analyzed an Ising-like model with a majority rule dynamics under the action of noise and an external modulation, interpreted as an opinion model by imitation under a fashion wave or propaganda, and found an SR that depends on the randomness of the small-world network. In Gimenez et al. [24,25] the authors found an SR phenomena in a variation of the Sznajd model subject to the combined effects of a stochastic driving and a periodic signal.

In this article, we investigate the CVM under the influence of an oscillating field, with the aim of broadening our understanding of the SR phenomenon in opinion dynamics models. Although the SR was already observed in the Galam, Sznajd and majority rule models among others, the ordering properties of these models are very different in nature to those of the VM studied in this article. This is due to the fact that the ordering dynamics of these “non-linear” models in mean-field is well described by a Ginzburg–Landau equation with a double-well potential like in the Ising model, while in the VM the associated potential is zero (zero drift), and thus the dynamics is driven only by noise [49]. A consequence of this difference is that the non-linear models have an order–disorder transition at a finite probability p>0 of contrarian behavior, while in the CVM the transition is at a $p^{*}=1/\left(N+1\right)$ as mentioned above, which vanishes in the thermodynamic limit. This peculiar behavior of the CVM is closely related to the zero-noise transition that exhibits the two-state noisy VM [50,51] as well as the noisy multi-state VM [52,53,54,55] where, besides imitation, voters can spontaneously switch state.

The article is organized as follows. In Section 2, we describe the model and the topology of interactions among agents, and we define the magnitudes that we use to characterize the system. In Section 3, we present the numerical results and develop an MF approach. Finally, in Section 4 we summarize the results and draw some conclusions.

## 2. Methods

### 2.1. The Model

We consider a population of *N* interacting agents that can hold one of two possible opinions S=1 or S=−1 on a given issue, for instance, in favor (opinion S=+1) or against (opinion S=−1) marijuana legalization. Each agent interacts with its neighbors in a given topology, and can either adopt the opinion of a neighbor (alignment/voter) or take its opposite opinion (anti-alignment/contrarian). The dynamics is defined as follows. In a time step Δt=1/N, two neighboring agents *i* and *j* with opinions $S_{i}$ and $S_{j}$, respectively, are randomly chosen. Then, *i* copies *j*’s opinion ($S_{i}\to S_{i}=S_{j}$) with probability $p^{j}$, or adopts the opposite opinion of agent *j* ($S_{i}\to S_{i}=-S_{j}$) with the complementary probability $1-p^{j}$. We note that this definition of the probabilities for the voter ($p^{j}$) and contrarian ($1-p^{j}$) behavior is analogous to that used in related models [24,25], but opposite to the definition used in the CVM [27] introduced in Section 1. The copying probability $p^{j}$ depends on $S_{j}$ and a parameter T≥0, and varies with time according to an oscillating periodic field $H\left(t\right)=H_{0}\sin\left(wt\right)$ that represents an external propaganda:(1)$$\begin{array}{c} p^{j}\left(t\right)=\frac{e^{[1+S_{j}H\left(t\right)]/T}}{e^{[1+S_{j}H\left(t\right)]/T}+e^{-[1+S_{j}H\left(t\right)]/T}}. \end{array}$$
Here, $H_{0}$ ($0\leq H_{0}\leq 1$) is the amplitude of the applied field and ω=2π/τ is the angular frequency of the field, defined by its period τ. We can see that $p^{j}$ is larger when $S_{j}$ is aligned with *H* [$\mathrm{sign}\left(S_{j}\right)=\mathrm{sign}\left(H\right)$], which models a situation where agents are more likely to adopt the opinion of individuals that are aligned with the propaganda. Therefore, the model assumes that it is easier to trust individuals whose opinions are in line with an external propaganda than individuals that have an opinion opposite to the propaganda. Furthermore, $p^{j}$ becomes larger as the intensity of the field $H_{0}$ increases. The parameter *T* controls the ratio $p_{j}/\left(1-p_{j}\right)$ between the probability to have an imitation and a contrarian behavior in a given iteration, respectively. In the T→0 limit, $p_{j}$ approaches 1, and thus only the imitation mechanism is present, which is equivalent to the original voter model (VM) dynamics. In the opposite limit, T→∞, $p_{j}$ approaches 1/2, and thus agent *i* adopts the state $S_{i}=1$ or $S_{i}=-1$ at random, independently of $S_{i}$. This corresponds to a situation where the system is driven purely by noise, and thus agents take a random state in each interaction. Therefore, we see that the parameter *T* plays a role analogous to that of the temperature in thermodynamic systems, and is proportional to the level of external noise. For this reason, *T* is called *social temperature*. We note that in the T→0 ($p_{j}\to 1$) and T→∞ ($p_{j}\to 1/2$) limits, the dynamics becomes independent of the field.

### 2.2. Topology of Interactions

We considered five different topologies that represent the structure of interactions— connections—between the agents, as we describe below.

**Complete graph (CG):** This is the simplest topology in which every agent is connected to every other agent in the population (all-to-all interactions).**Lattices:** Agents are located at the sites of a lattice with periodic boundary conditions, and interact with their first nearest-neighbors (NNs). The number of neighbors depends on the lattice: one-dimensional (1D) lattice or ring (two NNs), two-dimensional (2D) square lattice (four NNs), 2D triangular lattice (six NNs) and 2D hexagonal lattice (three NNs).

### 2.3. Magnetization and Signal-to-Noise Ratio

The behavior of the system at the macroscopic level is well described by the mean opinion or magnetization *m*, defined as the average value of the opinion over all agents in the population:(2)$$m\equiv \frac{1}{N}\sum_{i=1}^{N}S_{i},$$
which varies between −1 (opinion −1 consensus) and 1 (opinion +1 consensus). The m=0 case corresponds to a perfectly polarized population with an equal number N/2 of + and − agents.

Given that it is more likely to adopt an opinion that is aligned with the propaganda, we shall see that *m* tends to be positive when *H* is positive, and vice versa. Therefore, we expect that *m* responds to the external field, trying to follow its periodic oscillations. However, these oscillations could be affected by fluctuations in *m* originating from the external noise controlled by *T*, and also by the population noise (of order $N^{-1/2}$), intrinsic of finite-size systems. Then, as we are interested in studying the combined effects of external forcing and noise over *m*, we shall study the signal-to-noise ratio (SNR), a measure that compares the level of a desired signal to the level of the background noise. The SNR is defined as the difference between the signal power S and the noise power N in the Fourier space F(Ω), relative to N,
(3)$$SNR\equiv \frac{S-N}{N}.$$
Here, S is calculated as the height of the peak of F(Ω), which takes place at the frequency corresponding to the external field [F(ω)], and N denotes the estimated level of the background noise, calculated as the average value of F(Ω) outside the peak. An SNR>0 indicates that the signal is stronger than the background noise, and thus *m* shows a clear response to the field, whereas the case SNR≃0 corresponds to a very weak or none response.

## 3. Results

We run extensive Monte Carlo simulations of the model in the five topologies described in Section 2.2, for fields of amplitudes $H_{0}=0.02$, 0.01 and 0.5, periods τ=128, 256, 512 and 1024, and several temperatures. Initially, each agent takes the opinion state S=1 or S=−1 with the same probability 1/2, and thus m(0)≃0. We measured the time evolution of the magnetization *m*, the signal-to-noise ratio, and the amplitude and lag of oscillations in *m*. The results are described in the following sections.

### 3.1. Evolution of the Magnetization

In Figure 1a, we show the time evolution of *m* in a single realization of the dynamics for a system composed by N=1000 agents on a complete graph (CG), under the influence of an external field *H* ($H_{0}=0.1$ and τ=512), and for different temperatures.

We see that the behavior of m(t) varies with the temperature, and allows to distinguish four different regimes. For T=0 (red curve) the system is equivalent to the original VM, in which each agent copies the opinion of a random neighbor with probability 1.0. As it is known from the VM dynamics [1,2], the magnetization is conserved at each time step, and thus *m* performs a symmetric random walk until it reaches one of the two absorbing states (m=−1 consensus in this case). When the temperature is increased to the value T=0.25 (green curve), the copying probability oscillates slightly below 1.0. Then, the anti-alignment dynamics acts as a small noise that removes the system from the absorbing states m=±1, and thus *m* fluctuates and stays close to the extreme values m=±1 until a large fluctuation leads the system from one absorbing state to the other, that is, *m* jumps from 1 to −1 and vice versa. This is reminiscent of the bimodal behavior observed in the noisy VM for small values of the noise [50,51]. For a larger temperature T=1.0 (blue curve), *m* exhibits noisy periodic oscillations that are coupled to the field (black curve), that is, with a frequency and a phase similar to those of *H*. Finally, for a very large temperature T=100 (orange curve), *m* displays fluctuations around m=0 (full disorder). This is a consequence of the fact that the copying probability is similar to 1/2 independent of *H* and the agent’s state, and thus every agent takes an opinion value S=1 or S=−1 at random, leading to a purely noisy dynamics.

Figure 1b shows the evolution of *m* in a single realization in the four different lattices described in Section 2.2. We observe that *m* has a behavior that is qualitatively similar to that described above for a CG. For small enough temperatures (see T=0.2), the response of the system is very low or none, and *m* does not follow the external field. In contrast, for higher temperatures the response is high, and *m* exhibits oscillations that are coupled to those of the field. Within this oscillatory regime we observe a more complex behavior, which is also present in CG. The amplitude of oscillations becomes smaller as the temperature increases, from T=0.5 to T=2.0 to T=5.0. We shall explore this behavior in more detail in Section 3.5, and show that the amplitude is non-monotonic with *T*, with a maximum value at intermediate temperatures. These observations suggest that there is an optimal temperature value, or optimal noise, for the response of the system to the external signal; a *stochastic resonance* phenomenon also observed in related studies [24,25].

In Section 3.2, we try to give an insight into the transition between the bimodal and the oscillatory regimes, and in Section 3.3 we study the response of the system in more detail by means of the signal-to-noise ratio.

### 3.2. Transition Temperature

An estimation of the transition temperature $T_{c}$ between the bimodal regime for $T<T_{c}$ and the oscillatory regime for $T>T_{c}$ can be obtained by means of known results of the CVM dynamics [27] described in Section 1. At each time step, a voter and a neighbor are chosen at random. Then, the voter copies the state of its neighbor with probability 1−p (voter dynamics), or adopts its opposite state with the complementary probability *p* (contrarian behavior). In CG, it is shown in [27] that the distribution of the magnetization P(m) is peaked at m=0 (full disorder) for $p>p^{*}$, while P(m) exhibits two symmetric peaks at m=±1 (bimodal distribution) for $p<p^{*}$, where the transition value $p^{*}=1/\left(N+1\right)$ depends on the number of voters *N*. In order to make an analogy with our model, we associate the time average imitation probability ${\left(1+e^{-2/T}\right)}^{-1}$ of our model, obtained by setting H=0 in Equation (Equation 1), with the imitation probability 1−p of the CVM. We then expect a transition at a temperature $T_{c}$ for which ${\left(1+e^{-2/T_{c}}\right)}^{-1}≃1-p^{*}=1-1/\left(N+1\right)$, from where we obtain the estimation $T_{c}≃2/\ln N$.

The histogram of P(m) of our model for N=1000 agents reveals some features of the CVM without external field, and also new features (see Figure 2a). The system displays a transition from a bimodal behavior [two peaks in P(m)] for $T=0.25<T_{c}$ (top-left panel) to an oscillatory behavior for $T=1.0>T_{c}$ (bottom-left panel), where P(m) is symmetric around m=0 and shows two maximum values corresponding to the time oscillations of m(t) (blue curve in Figure 1a). In the top-right panel of Figure 2a, we see that P(m) is nearly uniform at $T_{c}≃2/\ln N≃0.2895$ for N=1000, showing that it is a fairly good estimation of the transition point. Finally, P(m) is symmetric and peaked at m=0 at the large temperature T=100≫1 (bottom-right panel).

These regimes are also characterized by the residence times $t_{r}$, i.e., the time interval between two consecutive changes of the sign of *m*. The histogram of the residence times is shown in Figure 2b for different temperatures. We see that for $T=0.25<T_{c}$ (top-left panel) the residence time exhibits a power–law distribution with an exponent similar to −3/2 (dashed line). This behavior can be explained from the first-passage time properties of a one-dimensional symmetric random walk. That is, for low *T* the system behaves approximately as the standard VM without external field, given that $p_{+}≃p_{-}≃1$. Thus, the evolution of *m* in a single realization can be approximated as that of a symmetric random walk in the interval [−1,1], due to the absence of the bias introduced by the field. Then, the distribution of time intervals between two consecutive crossings of the origin m=0 should follow the scaling $t_{r}^{-3/2}$ for large $t_{r}$, corresponding to the probability distribution of the first-passage time to the origin of a one-dimensional symmetric random walk (see for instance [56]). This power–law behavior is also very different from that observed in similar systems under the majority rule dynamics [42], where the distribution shows several peaks at the values $t_{r}=n\tau /2$ (n=1,2,…) that agree with the times at which the field changes sign. For moderate temperatures above $T_{c}$ (T=0.5 and 1.0), a peak appears at a value slightly smaller than τ/2=256, as it is expected from the fact that *m* stays synchronized with *H*. Finally, at a very large temperature T=100, the distribution decays exponentially with $t_{r}$ (bottom-right panel).

### 3.3. Signal-to-Noise Ratio

Figure 3 shows the signal-to-noise ratio (SNR) as a function of the temperature *T*, for an external field of amplitude $H_{0}=0.1$. Each curve corresponds to one of the five different interaction topologies described in Section 2.2. The system sizes are N=1024 for CG and 1D lattice, and N=32×32 for the 2D lattices (square, triangular and hexagonal). Panels correspond to different periods of the field (τ=128,256,512 and 1024), as indicated in the legends. Each curve shows a maximum at a given temperature $T^{*}$, indicating that there is an optimal temperature for the response of the system to the external field, as expected according to the stochastic resonance phenomenon. The resonance temperature $T^{*}$ does not seem to depend much on the period τ, but it does show a visible dependence on the topology or, more specifically, on its dimension: 1D, 2D and CG (mean-field or infinite dimension). We can see that $T^{*}$ increases as the dimension increases ($T_{1D}^{*}<T_{2D}^{*}<T_{CG}^{*}$), and that the height of the maximum decreases with the dimension [$SNR\left(T_{1D}^{*}\right)>SNR\left(T_{2D}^{*}\right)>SNR\left(T_{CG}^{*}\right)$]. Although the curves for the three different 2D lattices are quite similar, it seems that the position of the maximum $T^{*}$ slightly increases with the number of neighbors *z* [hexagonal (z=3) → square (z=4) → triangular (z=6)], while its value $SNR(T^{*})$ decreases with *z*. This tendency is in line with the 1D (z=2) and CG (z=∞) curves. These observations suggest that the response of the system becomes higher as the number of neighbors decreases.

The top panels of Figure 4 show the behavior of the SNR as a function of the field amplitude $H_{0}$ for five different temperatures (T=0.2, 0.3, 0.5, 1.3 and 2.3), on a CG of N=1024 agents (left panel), a 2D square lattice of N=32×32 sites (central panel), and a 1D lattice of N=1024 sites (right panel). The period of the field is τ=128. We can see that the SNR increases monotonically with $H_{0}$, as it is expected from the fact that a higher field intensity should lead to a higher response of the system.

In the bottom panels of Figure 4, we show the SNR as function of *T* for the corresponding topologies and parameters of the top panels. Each curve corresponds to a different value of $H_{0}$, from the relatively small value $H_{0}=0.6$ (black curve) to the highest possible value $H_{0}=1.0$ (magenta curve). As it happens for the $H_{0}=0.1$ case (see Figure 3), for $H_{0}=0.6$ and $H_{0}=0.7$ the SNR exhibits a single maximum at a resonance temperature $T^{*}$. For higher fields (T=0.8, 0.9 and 1.0), a second peak in the SNR appears at a lower temperature $T^{**}<T^{*}$, which becomes more pronounced as $H_{0}$ increases (the peak for $H_{0}=1.0$ is out of the shown scale). A similar behavior was observed in [24], and it was interpreted as a “double resonance” phenomenon. The location $T^{*}$ of the first peak does not seem to vary much with $H_{0}$, that is, the resonance temperature is $T^{*}≃1.3$ for all values of $H_{0}$ in the CG case, while $T^{*}≃1.0$ for the 1D lattice and the 2D square lattice, and all have values of $H_{0}$.

We have also measured the SNR on 2D triangular and 2D hexagonal lattices (plots not shown), and observed that the behavior of the SNR is very similar to that of the 2D square lattice shown in Figure 4.

Finally, to quantify the overall response of the system for a given topology and field, we defined the total response R as the area under the SNR vs. *T* curve. In Figure 5, we plot R vs. $H_{0}$ for the different topologies. The inset shows that the response increases quadratically with the amplitude of the field, $R∼H_{0}^{2}$. In addition, we notice from the main panel that R becomes larger as the dimension of the topology decreases, from CG to 2D to 1D, as we showed it happens for the peak of the SNR (Figure 3). These results imply that the response of the system is higher for low dimensional systems.

### 3.4. Amplitude and Lag of Mean Opinion Oscillations

In order to explore some properties of the oscillatory regime, we ran MC simulations on a CG of N=100 agents and calculated the average value of the magnetization, 〈m〉, over $10^{5}$ independent realizations of the dynamics. The time evolution of 〈m〉 is shown by symbols in Figure 6a for $H_{0}=0.1$, τ=512, and various temperatures.

We can see that the amplitude of oscillations *A* depends on the temperature *T* in a non-monotonic fashion, that is, *A* increases with *T* for low *T* values (T=0.2 and 0.5), and decreases with *T* for larger values (T=1.0,2.0 and 5.0). Therefore, *A* seems to exhibit a maximum at an intermediate temperature. This can be seen when we calculate the amplitude of oscillations as the absolute value of the difference between two consecutive extremes of 〈m(t)〉, i.e, $A\equiv \frac{1}{2}\left|{\langle m\rangle }_{max}-{\langle m\rangle }_{min}\right|$, averaged over the entire time range 0≤t≤2000. Here ${\langle m\rangle }_{max}$ and ${\langle m\rangle }_{min}$ are the maximum and minimum values of 〈m〉 in a given oscillation, respectively. The inset of Figure 6b shows *A* vs. *T* for $H_{0}=0.1$ and three different periods τ. For T≳0.7, the curve A(T) is independent of τ, while A(T) increases with τ for smaller temperatures. A possible explanation of this behavior is the following. When the field *H* is positive, the system has a bias ($p_{+}-p_{-}$) towards + opinions because $p_{+}>p_{-}$, and thus *m* tends to increase to positive values until *H* becomes negative and *m* starts to decrease (and analogously for a negative field). Then, given that the field remains positive or negative for a time period τ/2, increasing τ results in a bias that is sustained for a longer time and, consequently, *m* reaches a higher amplitude and thus *A* increases with τ. Besides, for a fixed temperature, when the period becomes large enough *m* reaches a maximum (or minimum) possible value that depends only on $H_{0}$ and, therefore, the amplitude *A* becomes independent of τ (*A* saturates with τ). This effect is observed in the inset of Figure 6b for T≥0.7, while for low enough temperatures the bias is small (both $p_{+}$ and $p_{-}$ are close to 1.0), and thus *m* reaches low amplitudes that eventually vanish as T→0 for a finite τ. We shall give an analytical insight of this observation in Section 3.5.

Figure 7a shows 〈m〉 vs. *t* for τ=512, temperature T=0.5, and three different field amplitudes $H_{0}$. In the inset of Figure 7b we plot the amplitude *A* vs. *T* for these fields. We observe that *A* is proportional to $H_{0}$, as it is expected from the fact that a more intense field should lead to a higher amplitude of *m*. In Section 3.5, we show that *A* increases linearly with $H_{0}$ for $H_{0}\ll T$.

Another magnitude that characterizes the oscillatory behavior of m(t) is the lag with respect to the external field, defined as the difference $L\equiv {\hat{t}}_{n}-{\tilde{t}}_{n}$ between the times ${\hat{t}}_{n}$ (n=0,1,2,…) where m(t) reaches its extreme values (maximum and minimum), and the corresponding times
(4)$${\tilde{t}}_{n}=\frac{(2n+1)\pi }{2\omega }$$
where the field $H\left(t\right)=H_{0}\sin\left(\omega t\right)$ reaches its extreme values. In Figure 6a we can see that, at a glance, the lag decreases as *T* increases. To explore this in more detail, we estimated the times ${\tilde{t}}_{n}$ of the extremes of 〈m〉, and calculated the differences with the extreme times ${\hat{t}}_{n}$ of H(t). The mean value of these differences over the time range 0≤t≤2000 was taken as the average lag 〈L〉.

In Figure 6b, we plot the lag with respect to the period as a function of *T*, for three different periods. We observe that the lag decreases with *T* from a value similar to τ/4 for low *T*, and seems to vanish as *T* increases. Furthermore, when we compare curves for different periods we can see that the relative lag L/τ decreases as the period increases. This is due to the fact that for a higher period the field changes more slowly with time, allowing the system to better follow the external signal, which translates into a magnetization that oscillates with a phase that is closer to that of *H*.

When we take a look at Figure 7a, we can see that the lag does not seem to depend much on $H_{0}$. In Figure 7b, we plot L/τ vs. *T* for τ=512 and for the same fields as panel (a). We observe a behavior with *T* analogous to that of Figure 6b. We also see that the lag seems quite independent of the field for low $H_{0}$ ($H_{0}=0.02$, 0.10), but it becomes smaller for larger values of $H_{0}$ ($H_{0}=0.5$).

In Section 3.5, we develop a mean-field approach that gives an analytical insight into these results.

### 3.5. Mean-Field Approach

The evolution of the fraction of agents with opinion +1, $\sigma_{+}$, in the N→∞ limit is given by the following rate equation:(5)$$\begin{array}{c} \frac{d\sigma_{+}}{dt}=\left[1-p_{-}\left(t\right)\right]\sigma_{-}^{2}+\left[p_{+}\left(t\right)-p_{-}\left(t\right)\right]\sigma_{+}\sigma_{-}-\left[1-p_{+}\left(t\right)\right]\sigma_{+}^{2}, \end{array}$$
where
(6)$$\begin{array}{c} p_{+}\left(t\right)=\frac{e^{\frac{1+H(t)}{T}}}{e^{\frac{1+H(t)}{T}}+e^{-\frac{1+H(t)}{T}}}\;\;\;and\;\;\;p_{-}\left(t\right)=\frac{e^{\frac{1-H(t)}{T}}}{e^{\frac{1-H(t)}{T}}+e^{-\frac{1-H(t)}{T}}} \end{array}$$
are the probabilities that an agent *i* copies the state $S_{j}=+1$ and $S_{j}=-1$, respectively, of a neighboring agent *j*, as defined in Equation (Equation 1). Furthermore, $\sigma_{-}=1-\sigma_{+}$ is the fraction of −1 agents. The first two terms in Equation (Equation 5) correspond to the transition $S_{i}=-1\to S_{i}=+1$, leading to a gain of 1/N in $\sigma_{+}$. The first term represents an interaction where two −1 agents are chosen at random with probability $\sigma_{-}^{2}$, and then one adopts the opposite opinion of its neighbor with probability $1-p_{-}$, while the second term corresponds to a −1 agent adopting the opinion of a +1 agent with probability $p_{+}$. Analogously, the third and fourth terms correspond to a loss of 1/N in $\sigma_{+}$, due to the transition $S_{i}=+1\to S_{i}=-1$, where a +1 agent adopts the opinion of a −1 agent with probability $p_{-}$ (third term), and the opposite opinion of a +1 agent with probability $1-p_{+}$ (fourth term). From Equation (Equation 5), the evolution of the magnetization $m=\sigma_{+}-\sigma_{-}$ is given by
(7)$$\begin{array}{c} \frac{dm}{dt}=\left(p_{+}+p_{-}-2\right)m+p_{+}-p_{-}, \end{array}$$
where $p_{+}$ and $p_{-}$ are short notations for $p_{+}\left(t\right)$ and $p_{-}\left(t\right)$. Although an exact analytical solution of Equation (Equation 7) is difficult to obtain for general parameter values, we can still obtain an approximate solution in the limit of a small external field as compared to the temperature. Then, we arrive at the following approximate expression for the evolution of the magnetization in the $H_{0}\ll T$ limit (see Appendix A for calculation details):(8)$$\begin{array}{c} m\left(t\right)≃m_{0}\;e^{\alpha t}+\frac{H_{0}\;\alpha \left(2+\alpha \right)\left[\ln(2+\alpha )-\ln(-\alpha )\right]\left\{\omega \left[\cos\left(\omega t\right)-e^{\alpha t}\right]+\alpha \sin\left(\omega t\right)\right\}}{2(\alpha^{2}+\omega^{2})}, \end{array}$$
where
(9)$$\begin{array}{c} \alpha \equiv -\frac{2}{1+e^{2/T}} \end{array}$$
is a constant parameter that depends on *T*, and $m_{0}=m\left(t=0\right)$ is the initial magnetization. In Figure 6a and Figure 7a, we compare the approximation from Equation (Equation 8) (solid lines) with the average magnetization 〈m〉 obtained from MC simulations on a CG of N=100 agents (symbols), for several temperatures. Curves in Figure 6a correspond to a field of small amplitude $H_{0}=0.1$. We observe a good agreement for all values of *T* even though, as expected, the approximation improves as *T* increases, with an estimated relative error that is less than 4% for T≥0.5. The reason for this good agreement is because we are simulating temperatures larger than $H_{0}=0.1$, where the analytical approximation is valid, given that for T<0.1 the copying probabilities $p_{+}$ and $p_{-}$ are approximately 0.99999999, which are taken numerically as 1.0 in the simulations within the machine precision, corresponding to the original VM (zero external field). However, as we see in Figure 7a for T=0.5, the analytical approximation works well for the small field amplitudes $H_{0}=0.02$ and 0.1 (squares and circles), but does not correctly capture the oscillations in *m* for the $H_{0}=0.5=T$ case (diamonds vs. blue solid line). In order to check that this discrepancy comes from the analytical approximation Equation (Equation 8) that works well only in the $H_{0}\ll T$ limit, we have numerically integrated Equations (Equation 16) and (A2) that give the exact analytical solution of the rate Equation (Equation 7). The pink solid line in Figure 7a corresponds to $H_{0}=0.5$, where we see a very good agreement with MC results (diamonds). This shows the validity of Equation (Equation 7) for all values of $H_{0}$.

We can see from Equation (Equation 8) that the two terms proportional to $e^{\alpha t}$ shift the center of oscillations to a value that depends on $m_{0}$, α and ω. As α<0 for all T≥0, this shift decays to zero exponentially fast. For low *T* (low α) this decay is slow, and oscillations remain off-centered for a long time [see for instance the T=0.2 curve in Figure 6a]. Eventually, in the long time limit, m(t) oscillates around m=0 with frequency ω [see the T≥0.5 curves in Figure 6a].

It becomes instructive to estimate how the amplitude of oscillations *A* behaves with *T* for general values of the parameters $H_{0}$ and ω from the approximate expression Equation (Equation 8). For this, it is better to work in the long time limit, where
(10)$$\begin{array}{c} m\left(t\right)≃\frac{H_{0}\;\alpha \left(2+\alpha \right)\left[\ln(2+\alpha )-\ln(-\alpha )\right]\left\{\omega \cos(\omega t)+\alpha \sin(\omega t)\right\}}{2(\alpha^{2}+\omega^{2})}, \end{array}$$
and thus oscillations are centered at m=0. Then, by setting the time derivative of m(t) from Equation (Equation 10) to zero (extreme value of *m*), we obtain the relation
(11)$$\begin{array}{c} \cos\left(\omega \;{\hat{t}}_{n}\;\right)=\frac{\omega }{\alpha }\sin\left(\omega \;{\hat{t}}_{n}\;\right), \end{array}$$
at the times
(12)$${\hat{t}}_{n}=\frac{1}{\omega }\tan^{-1}\left(\frac{\alpha }{\omega }\right)+\frac{(n+1)\pi }{\omega }$$
where *m* reaches its maximum (n=0,2,4,…) and minimum (n=1,3,…) values in a given oscillation. Then, the amplitude $A=\frac{1}{2}\left|m\left({\hat{t}}_{n+1}\right)-m\left({\hat{t}}_{n}\right)\right|$ can be obtained by replacing the equivalence from Equation (Equation 11) for ${\hat{t}}_{n+1}$ and ${\hat{t}}_{n}$ in Equation (Equation 10), which leads to
$$\begin{array}{c} A≃\frac{1}{2}H_{0}\;\left(2+\alpha \right)\left[\ln(2+\alpha )-\ln(-\alpha )\right]|\sin\left[\tan^{-1}\left(\alpha /\omega \right)\right]|, \end{array}$$
or, in terms of the period τ=2π/ω and the temperature *T* is
(13)$$A≃\frac{2H_{0}}{T(1+e^{-2/T})}|\sin\left\{\tan^{-1}\left[-\frac{\tau }{\pi (1+e^{2/T})}\right]\right\}|\;\;\;\mathrm{for}\;H_{0}\ll T.$$
The approximate expression for *A* from Equation (Equation 13) is compared with MC simulations (symbols) in the inset of Figure 6b for $H_{0}=0.1$ and various values of τ (solid lines), and in the inset of Figure 7b for τ=512 and various values of $H_{0}$ (solid lines). As expected, the approximation works well for the $H_{0}=0.1$ case (Figure 6b), but it fails for the $H_{0}=0.5$ case of Figure 7b, where the predictive value of *A* exceeds 1.0 for intermediate temperatures.

From Equation (Equation 13) we can see that *A* increases linearly with $H_{0}$ (Figure 7b). We can also see that *A* saturates to a given value as τ increases, for a fixed temperature *T* [dashed line in the inset of Figure 6b]. This saturation value can be calculated from Equation (Equation 7) assuming that for a very long period τ≫1 the field *H* varies so slowly in time that $p_{+}$ and $p_{-}$ can be approximated as constant parameters, where *m* reaches the quasi-stationary value $m^{s}=\left(p_{+}-p_{-}\right)/\left(2-p_{+}-p_{-}\right)$ from Equation (Equation 7) (we note that $m^{s}$ agrees with the expression obtained in [41] for a similar model subject to a constant field). Then |m| reaches its maximum quasi-stationary value $|m_{max}^{s}|=\left(e^{2H_{0}/T}-e^{2H_{0}/T}\right)/\left(e^{2H_{0}/T}+e^{-2H_{0}/T}+2e^{-2/T}\right)$ for the extreme field value $\left|H\right|=H_{0}$, which is shown by a dashed line in the inset of Figure 6b. The saturation value $|m_{max}^{s}|$ increases as *T* decreases, and approaches 1.0 as T→0 (out of the shown scale). Then, for small temperatures $|m_{max}^{s}|$ is high, but *m* cannot reach this maximum possible amplitude for a low value of τ (τ≤1024 in simulations) because $p_{+}≃p_{-}≃1$, and thus the bias in *m* given by $p_{+}-p_{-}$ is very small. For large temperatures the bias is also low because $p_{+}≃p_{-}≳0.5$, but $|m_{max}^{s}|$ is small, and thus *m* is able to reach this extreme value $|m_{max}^{s}|$. This is consistent with the observation in Section 3.4 that the amplitude *A* of *m* is small for low *T*, and becomes independent of τ for large T.

An approximate expression for the lag $L={\hat{t}}_{n}-{\tilde{t}}_{n}$ in the long time limit can be obtained by using the approximate expression for ${\hat{t}}_{n}$ from Equation (Equation 12) and ${\tilde{t}}_{n}$ from Equation (Equation 4). We recall that ${\hat{t}}_{n}$ and ${\tilde{t}}_{n}$ (n=0,1,2,⋯) are the times at which *m* and *H* reach their respective extreme values. This leads to
(14)$$L≃\frac{1}{\omega }\tan^{-1}\left(\frac{\alpha }{\omega }\right)+\frac{\pi }{2\omega }\;\;\;\mathrm{for}\;H_{0}\ll T.$$
Then, the lag with respect to the period τ in terms of *T* and τ becomes
(15)$$\frac{L}{\tau }≃\frac{1}{4}+\frac{1}{2\pi }\tan^{-1}\left[-\frac{\tau }{\pi \left(1+e^{2/T}\right)}\right]\;\;\;\mathrm{for}\;H_{0}\ll T.$$
The approximation from Equation (Equation 15) is plotted by solid lines in Figure 6b for different periods τ, and in Figure 7b for different field amplitudes $H_{0}$. We see a good agreement with MC simulations (symbols) for the entire range of temperatures except for the $H_{0}=0.5$ case. We also note from Equation (Equation 15) that the lag becomes independent of $H_{0}$ for small fields, as already mentioned in Section 3.4.

Finally, we can check from Equation (Equation 15) that L/τ→1/4 in the low temperature limit T→0, and that $L/\tau \to \frac{1}{2\pi }\tan^{-1}\left(-\frac{\tau }{2\pi }\right)+1/4$ in the high temperature limit T≫1, which becomes $L/\tau =\tau^{-1}$ for τ≫2π. This result explains why L/τ decreases as τ increases (see Figure 6b), and also why the lag is much smaller than the period (L≃1≪τ) for high enough temperatures and for the values of τ used in MC simulations.

## 4. Discussion and Conclusions

We studied a binary-state opinion formation model that incorporates the competition between two opposite social mechanisms for the adoption of an opinion, an imitation mechanism by which agents adopt the opinion of a random neighbor (voter dynamics), and a contrarian behavior (anti-voter dynamics) in which agents take the opposite opinion of a neighbor. The relative rate between the voter and the contrarian behavior is controlled by a parameter T≥0 (the social temperature), and is coupled to an oscillating field of period τ that mimics the influence of an external oscillating propaganda on individuals’ decisions.

We simulated the model on a complete graph and several regular lattices, and found that the qualitative behavior is similar in all these interaction topologies. As the temperature *T* is varied, we observed that the system exhibits a quite rich variety of behaviors. For T=0, the dynamics is only opinion imitation (voter dynamics with zero noise), which eventually leads the system to an absorbing state of consensus in one of the two opinions; a property of the original voter model on all topologies [1,2]. For intermediate temperatures, the system exhibits two different phases separated by a transition temperature $T_{c}$. A bimodal phase for $T<T_{c}$ where the magnetization *m* (population’s mean opinion) in a single realization remains close to the extreme values m=+1 or m=−1, describing a population that shifts between ordered states, and an oscillatory phase for $T>T_{c}$ where *m* oscillates around 0, corresponding to a population that is easily influenced by the external propaganda, where individuals’ opinions oscillate over time following the mass media trends. Finally, for T≫1 the high level of noise drives the system to complete disorder, with a magnetization that fluctuates around m=0, corresponding to a stable opinion coexistence with similar fractions of agents holding one and the other opinion.

Within the bimodal phase for $T<T_{c}$, *m* fluctuates close to the consensus value m=+1 (m=−1) for long periods until it jumps to the opposite value m≃−1 (m≃+1), leading to a bimodal distribution in *m* that is peaked at m=±1, a well-known feature of the original noisy voter model [50] and its various versions. The time that it takes the system to cross the m=0 line (or residence time $t_{r}$) is distributed according to a decaying power law with an exponent similar to 3/2, which is associated to the first-passage properties of a one-dimensional symmetric random walk. This power-law decay is different from the behavior observed in related opinion models with majority rule dynamics [42], where the residence time distribution for low noise shows peaks at multiple values of τ/2. The oscillatory regime for $T>T_{c}$ exhibits various properties. The magnetization *m* oscillates with the same frequency as the field *H*, but with a lag respective to *H* that decreases monotonically with *T*. The amplitude of oscillations have a maximum at an intermediate temperature. An insight into these results was given by a mean-field approach that reproduces well the dynamics on an infinite large complete graph. At low temperatures, the system tends to follow the voter dynamics where *m* is conserved, thus the amplitude of oscillations is very small, whereas for high temperatures the noise is high and thus the system decouples from the field and is mainly driven by stochastic fluctuations, leading to small oscillations in *m* as well. At an optimal intermediate temperature, *m* is highly coupled to *H* and reaches its largest oscillations. This phenomenon, which can be seen in infinite as well as in finite systems, is closely related to another interesting phenomenon induced by the intrinsic stochasticity of finite systems that enhances the response to an external forcing, that is, the stochastic resonance. By using the signal-to-noise ratio (SNR) as a measure of the system’s response (*m*) to the external field, we found a resonance temperature $T^{*}$ at which the response reaches a maximum value. The resonance peak is higher and shifted towards lower temperatures as the dimension of the system decreases, from CG to 2D to 1D. Although the SNR vs. *T* curves are similar in 2D lattices, it appears to be a small dependency with the number of neighbors and, in general, we can say that the resonance peak increases and shifts towards lower temperatures as the number of neighbors decreases, from CG (N−1 NNs), to 2D triangular lattice (six NNs), to 2D square lattice (four NNs), to 2D hexagonal lattice (three NNs), to 1D lattice (two NNs). This means that the response of the system at resonance is intensified as the number of interacting neighbors is reduced: less neighbors induce a higher response. This result is in line with the behavior of the overall response of the system, measured by the area under the SNR vs. *T* curve, which increases as the number of neighbors decreases.

As a future work, we plan to investigate the dynamics of this model in complex networks, which would allow us to study how the stochastic resonance phenomenon is affected by the network’s degree distribution. In particular, we are interested in exploring the dependence of the resonance peak on the number of neighbors. It might also be worthwhile to explore the effects of an oscillating propaganda and its associated stochastic resonance in the contrarian voter model with the addition of an intermediate opinion state S=0 that could represent undecided or moderate agents. Finally, it might be interesting to apply this model to the study of data obtained from polls about the dynamics of ideological change and satisfaction with democracy, which is characterized by turns in the population’s ideology—from left to right and vice versa—over the years [57].

## Figures and Tables

**Figure 1 entropy-24-01140-f001:**
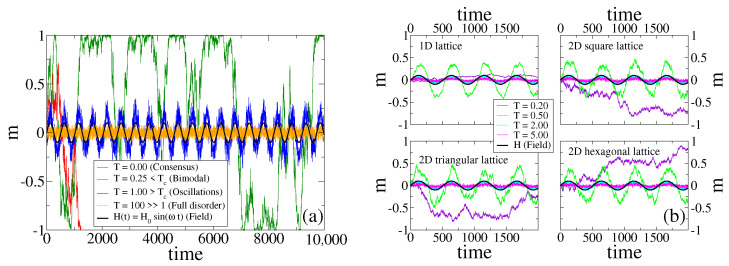
(**a**) Time evolution of the magnetization *m* in a single realization of the dynamics for a system of N=1000 agents on a complete graph, subject to a periodic field $H\left(t\right)=H_{0}\sin\left(\omega t\right)$ of amplitude $H_{0}=0.1$ and period τ=512 (ω=2π/τ), and an external noise associated to a temperature *T*. Each curve corresponds to a different temperature, as indicated in the legend. Four behaviors are observed: −1 consensus for T=0, bimodal behavior for $T=0.25<T_{c}$, oscillations for $T=1.0>T_{c}$, and full disorder for T=100≫1. Here, $T_{c}≃0.2895$ is the transition temperature. The field H(t) is also plotted for reference. (**b**) Evolution of *m* in a single realization for the same parameters as in panel (**a**), on a 1D-lattice of N=8192 sites (top-left), a 2D square lattice of size N=64×64 (top-right), a 2D triangular lattice of size N=64×64 (bottom-left), and a 2D hexagonal lattice of size N=64×64 (bottom-right).

**Figure 2 entropy-24-01140-f002:**
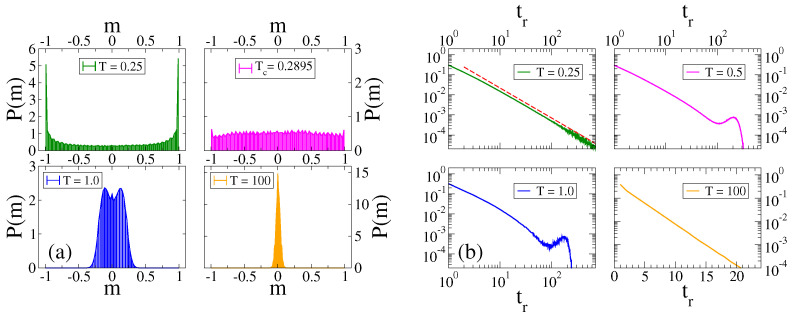
(**a**) Histogram of *m* for the same system and parameters as those in Figure 1, and for the temperatures indicated in the legends. At the transition point $T_{c}≃0.2895$, the distribution P(m) is uniform (top-right). (**b**) Normalized histograms of the residence time $t_{r}$ for the same system and parameters of panel. (**a**) Each plot corresponds to a different temperature *T*, as indicated in the legends. The bottom-right plot is in linear-log scale, while the other plots are in double logarithmic scale. The dashed line in the top-left plot has slope −3/2. Each histogram was obtained by running a single realization up to a time $10^{8}$.

**Figure 3 entropy-24-01140-f003:**
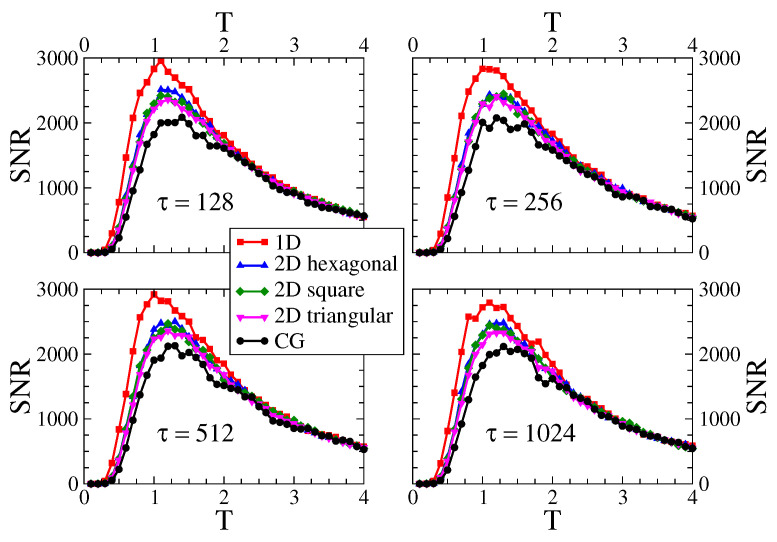
Signal-to-noise ratio SNR vs. temperature *T* on the five interaction topologies indicated in the legend. The system sizes are N=1024 for the 1D lattice and CG, and N=32×32 for the 2D lattices (hexagonal, square and triangular). The amplitude of the external field is $H_{0}=0.1$. Each panel corresponds to a different period τ=128, 256, 512 and 1024.

**Figure 4 entropy-24-01140-f004:**
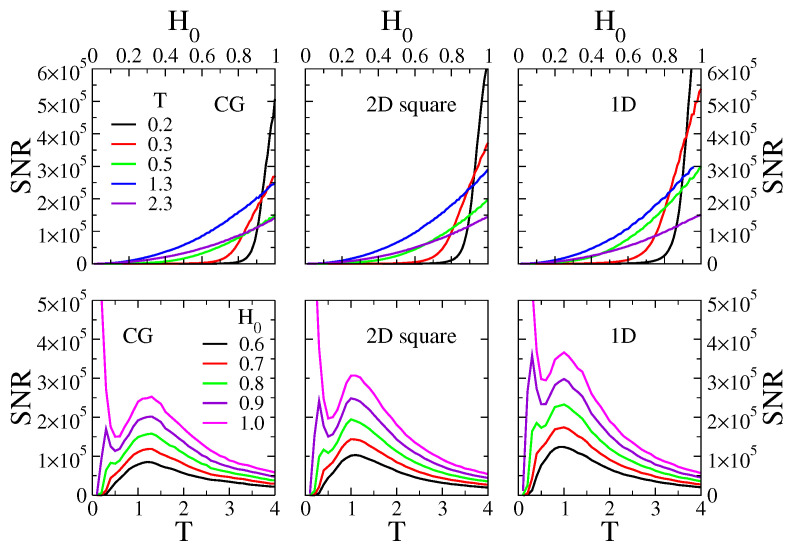
Top panels: signal-to-noise ratio SNR as a function of the amplitude of the external field $H_{0}$, for temperatures T=0.2,0.3,0.5,1.3 and 2.3. Bottom panels: SNR vs. *T* for $H_{0}=0.6,0.7,0.8,0.9$ and 1.0. The period of the field is τ=128. Left, central and right panels correspond to simulations on a complete graph (CG) of N=1024 nodes, a 2D square lattice of N=32×32 sites and a 1D lattice of N=1024 sites, respectively.

**Figure 5 entropy-24-01140-f005:**
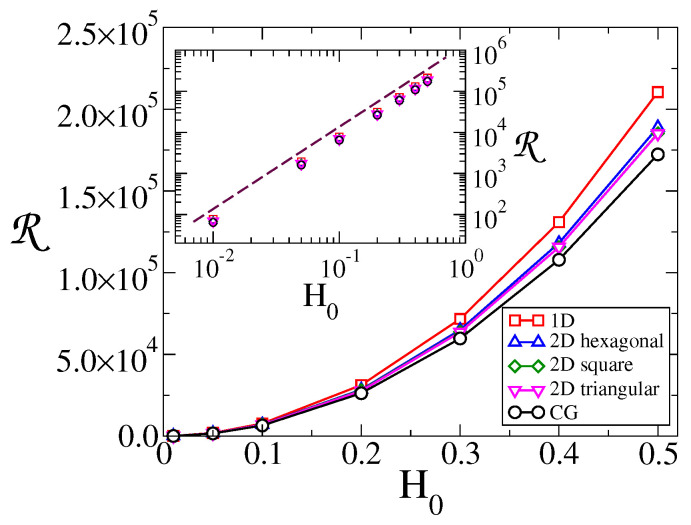
Total response R vs. field amplitude $H_{0}$, for the topologies indicated in the legend. The period of the field is τ=128, and the system sizes are N=1024 for CG and 1D lattice, and N=32×32 for 2D lattices. Inset: R vs. $H_{0}$ on a log-log scale. The dashed line has slope 2.

**Figure 6 entropy-24-01140-f006:**
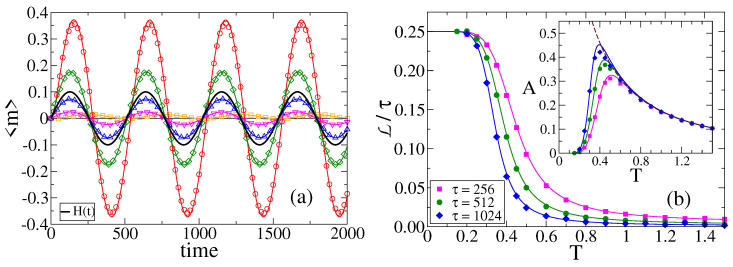
(**a**) Time evolution of the average magnetization 〈m〉 for temperatures T=0.2 (squares), T=0.5 (circles), T=1.0 (diamonds), T=2.0 (up triangles) and T=5.0 (down triangles), on a CG of size N=100, with a field of amplitude $H_{0}=0.1$ and period τ=512. The average was done over $10^{5}$ independent realizations of the dynamics. Solid lines are the analytical approximation from Equation (Equation 8). (**b**) Lag L respect to the period τ vs. temperature *T* for field periods τ=256 (squares), τ=512 (circles) and τ=1024 (diamonds), and amplitude $H_{0}=0.1$, for the same topology of panel (**a**). Solid lines are the approximation from Equation (Equation 15). Inset: Amplitude *A* vs. *T* for the same parameter values as in the main panel. Solid lines are the analytical approximation from Equation (Equation 13).

**Figure 7 entropy-24-01140-f007:**
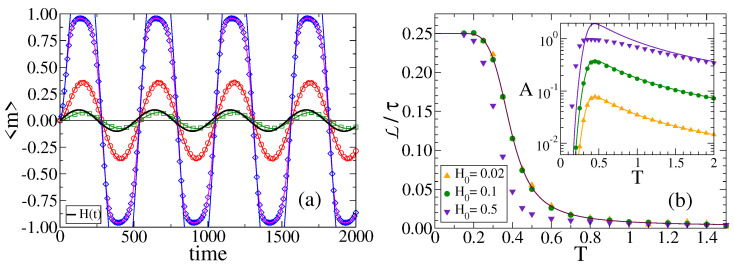
(**a**) Time evolution of the average magnetization 〈m〉 over $10^{5}$ realizations of the dynamics on a CG of size N=100, with temperature T=0.5, under a field of period τ=512 and amplitudes $H_{0}=0.02$ (squares), $H_{0}=0.1$ (circles) and $H_{0}=0.5$ (diamonds). The solid pink line corresponds to the numerical integration of Equations (Equation 16) and (A2), while the other solid lines are the analytical approximation from Equation (Equation 8). (**b**) Normalized lag L/τ vs. temperature *T* for a field of period τ=512 and amplitudes $H_{0}$ indicated in the legend. The solid line is the approximation from Equation (Equation 15). Inset: *A* vs. *T* for the same parameter values as in the main panel. Solid lines are the approximation from Equation (Equation 13).

## Data Availability

All relevant data are contained within the paper.

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
