# Peer review of "Contrarian Voter Model under the Influence of an Oscillating Propaganda: Consensus, Bimodal Behavior and Stochastic Resonance"

_entropy, 2022, doi:10.3390/e24081140_

Round 1

Reviewer 1 Report

In this paper the authors consider the Contrarian Voter Model introduced by Banisch (not to be confused with the Voter Model with Contrarians of Masuda) where the tendency to adopt the Contrarian attitude depends on time imitating the effect of an external influence, such as advertising, that favors periodically one of the two options. They do so by introducing the concept of "social temperature" T as a quantifier of the noise in the model, such that the probability to adopt the contrarian attitude depends on a particular form of this temperature. The authors find that the model exhibits the phenomenon of Stochastic Resonance by which the response, measured by the way the average opinion follows the periodic variation of the external influence, is optimal for a given value of the temperature. They study this phenomenon on a complete graph (every agent connected to every other agent) and on different regular topologies, proposing for future work the study of complex topologies typical of social systems.

I have found the paper very well written, giving full details about the modeling, simulation and analytical tools used. The results are thoroughly described (dependence on the temperature, external field, topology) and they are conveniently illustrated by the selected figures. Given our previous knowledge, one can not say that the existence of Stochastic Resonance is unexpected, but the analysis shows interesting features such as the dependence with the number of neighbors (the fewer, the more intense the resonance) which have relevance in the sociological interpretation of the model. 

As I say, the paper is well written and I do not have any major comments. My recommendation is to publish the paper as it is. Nevertheless, there are some points that the authors might optionally want to take into consideration.

-Do the authors have any explanation of the behavior of the residence time (power-law behavior)? Is a similar power-law dependence observed in other cases of stochastic resonance?

-There is no need to specify that |m_max-m_min|=|m_min-m_max| as it says in the definition of the amplitude A, unless there is some subtlety in the definition.

-I appreciate the need for analytical expressions, but it might be helpful to add the numerical solution of Eq. (7) simply performing numerically the integrals of (A1-A2) to compare with the simulation results and check the validity of the rate equation for large amplitude (diamonds in Figure 7). 

Reviewer 2 Report

Report on :

Contrarian Voter Model under the influence of an Oscillating
Propaganda: Consensus, Bi-stability and Stochastic Resonance

by M. Cecilia Gimenez, Luis Reinaudi  and Federico Vazquez

 The general impresion is good, and I consider that this   manuscript is adequate for publishing in Entropy. However, I consider that there are some points to be consider.

i) What is the origin of the SR spike at small T? In neural systems it is

   associated with spontaneous firing of neurons, but here?

ii) What is the criterion for adopting R as a total  response? This, in  

   "conventional" SR, is related to the total "power" delivered  to the 

    system. Please comment.

iii)  Qualitatively, what could be the effect of the existence of a group

      of undecided agents? Please comment.

iv)  Is it possible to think of an adjustment of  parameters that would

     make the election more favorable for one of the groups? Please

    comment.
